

# The Erlang distribution approximates the age distribution of incidence of childhood and young adulthood cancers

Aleksey V. Belikov, Alexey Vyatkin[†] and Sergey V. Leonov

Laboratory of Innovative Medicine, School of Biological and Medical Physics, Moscow Institute of Physics and Technology, Moscow, Russia
[†] Deceased.

## ABSTRACT

**Background**. It is widely believed that cancers develop upon acquiring a particular number of (epi) mutations in driver genes, but the law governing the kinetics of this process is not known. We have previously shown that the age distribution of incidence for the 20 most prevalent cancers of old age is best approximated by the Erlang probability distribution. The Erlang distribution describes the probability of several successive random events occurring by the given time according to the Poisson process, which allows an estimate for the number of critical driver events.

**Methods**. Here we employ a computational grid search method to find global parameter optima for five probability distributions on the CDC WONDER dataset of the age distribution of childhood and young adulthood cancer incidence.

**Results**. We show that the Erlang distribution is the only classical probability distribution we found that can adequately model the age distribution of incidence for all studied childhood and young adulthood cancers, in addition to cancers of old age.

**Conclusions**. This suggests that the Poisson process governs driver accumulation at any age and that the Erlang distribution can be used to determine the number of driver events for any cancer type. The Poisson process implies the fundamentally random timing of driver events and their constant average rate. As waiting times for the occurrence of the required number of driver events are counted in decades, and most cells do not live this long, it suggests that driver mutations accumulate silently in the longest-living dividing cells in the body—the stem cells.

Corresponding author
Aleksey V. Belikov,
belikov.research@gmail.com

## INTRODUCTION

Since the discovery of the connection between cancer and mutations in DNA, in the middle of the 20th century, there have been multiple attempts to deduce the law of driver mutation accumulation from the age distribution of cancer incidence or mortality (*Hornsby, Page & Tomlinson, 2007*). The proposed models, however, suffer from several serious drawbacks. For example, early models assume that cancer mortality increases with age according to the power law (*Nordling, 1953*; *Armitage & Doll, 1954*; *Knudson, 2001*), whilst already at that time it was known that many cancers display deceleration of mortality

growth at advanced age. Moreover, when large-scale incidence data have accumulated, it became clear that cancer incidence not only ceases to increase with age but, for at least some cancers, starts to decrease (*Saltzstein, Behling & Baergen, 1998*; *Harding, Pompei & Wilson, 2012*). More recent models of cancer progression are based on multiple unverified biological assumptions, consist of complex equations that incorporate many empirically determined parameter values, and still have not been shown to describe the decrease in cancer incidence at an advanced age (*Luebeck & Moolgavkar, 2002*; *Little & Wright, 2003*; *Michor, Iwasa & Nowak, 2006*; *Meza et al., 2008*; *Calabrese & Shibata, 2010*; *Luebeck et al., 2013*). It is also clear that an infinite number of such mechanistic models can be created and custom-tailored to fit any set of data, leading us to question their explanatory and predictive values.

Recently, we have proposed that the age distribution of cancer incidence is, in fact, a statistical distribution of probabilities for a required number of driver events to occur precisely by the given age, *i.e.*, a probability density function (*Belikov, 2017*). We define driver events as discrete molecular alterations that provide the cancer cell of origin and the resulting clone the necessary means to overcome both the intracellular and extracellular tumor suppressor mechanisms, such as cell cycle checkpoints, contact inhibition, immune surveillance, and promote growth in the absence of growth factors. Thus, our concept of a driver event integrates mutational, microenvironmental, immune and evolutionary perspectives (*Rozhok & De Gregori, 2015*; *Solary & Lapane, 2020*). It is also important to note that only the winner clone's evolutionary history determines the time of tumour appearance and thus the kinetics of cancer incidence; therefore, any mention of driver events in this article would always refer to driver events in the winner clone. The total amount of competing clones would rather affect the overall probability of having a cancer.

Previously, we tested the probability density functions of 16 well-known continuous probability distributions for fits with the CDC WONDER data on the age distribution of incidence for the 20 most prevalent cancers of old age (*Belikov, 2017*). The best fits were observed for the gamma distribution and its special case—the Erlang distribution, with the average $R^2$ of 0.995 (*Belikov, 2017*). Notably, these two distributions describe the probability of several successive independent random events occurring precisely by the given time. In fact, the Erlang distribution is the distribution of a sum of independent exponential variables, whereas the exponential distribution is the probability distribution of the time between events in a Poisson point process, *i.e.,* a process in which events occur continuously and independently at a constant average rate (*Birnbaum, 1954*). This allowed us to estimate the number of driver events, the average time interval between them and the maximal populational susceptibility, for each cancer type. The results showed high heterogeneity in all three parameters amongst the cancer types but high reproducibility between the years of observation (*Belikov, 2017*).

However, four other probability distributions—the extreme value (Gumbel), normal, logistic and Weibull—also showed good fits to the data, although inferior to the gamma and Erlang distributions. This leaves some uncertainty regarding the exceptionality of the gamma/Erlang distribution for the description of cancer incidence. Here we test these shortlisted distributions on the CDC WONDER data on childhood and young adulthood

**Table 1** Probability density functions and Python code for statistical distributions.

| Distribution | Probability density function | Python code |
|---|---|---|
| Gamma/Erlang | $f(t) = \frac{1}{b\Gamma(k)}\left(\frac{t}{b}\right)^{k-1}e^{-\frac{t}{b}}$ | def Erlang_pdf(k, b, t):return ( t**(k-1) * np.exp(-t/float(b)) ) / (b**k * gamma(k)) |
| Weibull | $f(t) = \frac{k}{b}\left(\frac{t}{b}\right)^{k-1}e^{-\left(\frac{t}{b}\right)^k}$ | def Weibull_pdf(k, b, t):return (k/b) * (t/b)**(k-1) * np.exp(-(t/b)**k) |
| Extreme value | $f(t) = \frac{1}{b}e^{-\left(\frac{t-\mu}{b}\right)}e^{-e^{-\left(\frac{t-\mu}{b}\right)}}$ | def Extreme_value_pdf(mu, b, t):return np.exp((mu - t) / b) * (1/b) * np.exp(-np.exp((mu - t) / b)) |
| Logistic | $f(t) = \frac{e^{\left(\frac{t-\mu}{b}\right)}}{b\left(1+e^{\left(\frac{t-\mu}{b}\right)}\right)^2}$ | def Logistic_pdf(mu, b, t):return (1/b) * np.exp((t - mu) / b) / np.square(1 + np.exp((t - mu) / b)) |
| Normal | $f(t) = \frac{1}{b\sqrt{2\pi}}e^{-\frac{1}{2}\left(\frac{t-\mu}{b}\right)^2}$ | def Normal_pdf(mu, b, t):return (1/(b * np.sqrt(2 * pi))) * np.exp(−0.5 * np.square((t - mu)/b)) |

cancers, using a grid search computational method to find global parameter optima. We show that the gamma/Erlang distribution is the only distribution that provides close fits for all tested cancers without involvement of negative age values. This result, taken together with our previous findings (*Belikov, 2017*), suggests that driver accumulation is governed by the Poisson process, both in childhood and in adult cancers. This is consistent with driver mutations that accumulate randomly, silently, at constant average rate and for many decades, likely in stem cells.

# METHODS

## Data acquisition

Data were collected as previously described in (*Belikov, 2017*). Briefly, United States Cancer Statistics Public Information Data: Incidence 1999–2012 were downloaded *via* CDC WONDER online database (http://wonder.cdc.gov/cancer-v2012.HTML). Results were grouped by 5-year Age Groups, Crude Rates were selected as an output and All Ages were selected in the Age Group box. The data were downloaded separately for each specific cancer type, upon its selection in the Childhood Cancers tab. Only cancers that show childhood/young adulthood incidence peaks and do not show middle/old age incidence peaks were analysed further. The middle age of each age group was used as the x value, *e.g.*, 17.5 for the "15–19 years" age group.

## Data analysis

For data analysis, custom scripts were written in Python 3.7, now publicly available at our GitHub repository: https://github.com/belikov-av/childhoodcancers. The following packages were used: Scikit-learn 0.22.1, Numpy 1.18.1, Scipy 1.4.1, Pandas 1.0.3, Plotly 4.5.4 and Matplotlib 3.1.3. Distributions were defined as shown in Table 1.

In order to find and visually demonstrate the globally optimal parameters for every distribution, a grid search method (*Jiménez, Lázaro & Dorronsoro, 2007*) was chosen, as opposed to standard maximum likelihood estimation methods that do not guarantee finding the global optimum. Grid search allowed us to explore all combinations of values

for two parameters ($k$ and $b$ for the Erlang and Weibull distributions; $\mu$ and $b$ for the extreme value, logistic and normal distributions) within the defined limits and with the defined steps (400 values for each parameter, 160000 grid nodes). For each pair of parameter values, the optimum value for the third parameter (amplitude, interpreted as the maximal populational susceptibility, see (*Belikov, 2017*)) was found using the golden section search method (*Pejic & Arsic, 2019*). In order to perform the golden section search, first the area under the incidence curve was estimated as the sum of actual incidence values for each age group. The left and right boundaries for the golden section search were then defined as an estimated area under the curve divided or multiplied by 100, respectively. After the optimal amplitude parameter value was found with the golden section search, the fitted distribution was generated and plotted alongside the incidence data and the goodness of fit ($R^2$) was calculated, for each node of the grid. Finally, the node with the highest $R^2$ was selected and its parameters deemed the global optimum.

## RESULTS

To test the universality and exceptionality of the gamma/Erlang distribution, the publicly available USA incidence data on childhood and young adulthood cancers were downloaded from the CDC WONDER database (see Methods). In addition to the gamma/Erlang distribution, the probability density functions of the following continuous probability distributions were selected for testing based on their good fits to adult cancers (*Belikov, 2017*): extreme value (Gumbel), logistic, normal and Weibull. We used a custom grid search script to explore all combinations of parameters and find global $R^2$ maxima (see Fig. 1 for gamma/Erlang distribution, Supplemental Information for other distributions and Methods for the detailed description). Results showed that the extreme value (Gumbel), logistic and normal distributions fit to some childhood cancers in a way that a large part of their probability density function appears in the negative range of $x$ values (Fig. 2). As the $x$ axis corresponds to patients' ages, this makes these three distributions uninterpretable in the biological context. Hence, they were excluded from the further analysis. The remaining gamma/Erlang and Weibull distributions are defined only for positive $x$ values, so there was no such problem (Fig. 2). The gamma/Erlang distribution provided closer fits to the childhood and young adulthood cancers than the Weibull distribution (Fig. 3, Table 2).

Importantly, the gamma distribution and the Erlang distribution derived from it are the only classical continuous probability distributions that describe the cumulative waiting time for $k$ successive random events, with the Erlang distribution differing only in counting events as integer numbers. Assuming this model describes the waiting time for real discrete random events such as driver mutations, the gamma/Erlang distribution provides the opportunity to get unique insights into the carcinogenesis process. We have previously proposed that the shape parameter $k$ of the gamma/Erlang distribution indicates the average number of driver events that need to occur in order for a cancer to develop to a stage that can be detected during clinical screening; the scale parameter $b$ indicates the average time interval (in years) between such events; and the amplitude parameter $A$ divided by 1,000 estimates the maximal susceptibility (in percent) of a given population to a given type of cancer (*Belikov, 2017*).

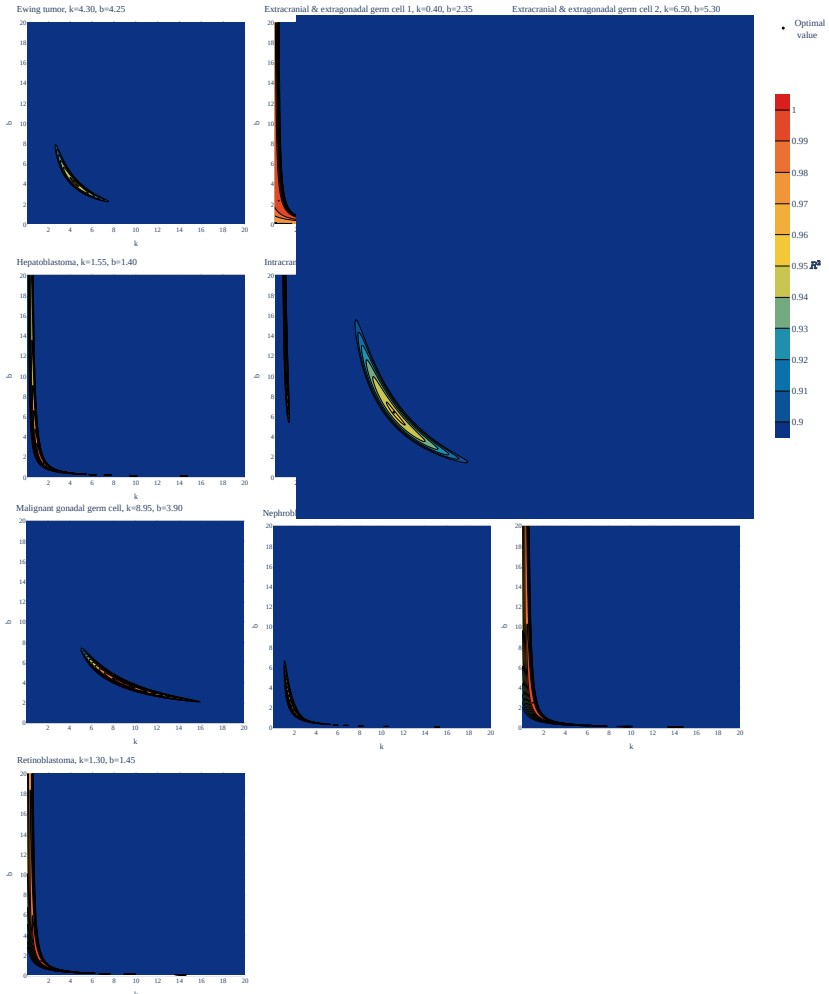

**Figure 1** **Goodness of fit of the gamma/Erlang distribution to the age distributions of incidence of 10 childhood/young adulthood cancer types as a function of various parameter combinations.** The $k$ and $b$ parameters were sampled with 0.05 interval, and for each pair the amplitude ($A$, not shown) and goodness of fit ($R^2$) were calculated. See Supplemental Information for the other distributions.

To obtain these parameter values, the gamma/Erlang distribution was fitted individually to incidence of each of 10 childhood/young adulthood cancer types (Fig. 3, Table 3).

The non-integer values of the shape parameter $k$ can be easily explained if we suppose that the studied population consists of the mixture of patients with slightly different numbers of driver events. The goodness of fit varied from 0.9495, for extracranial and extragonadal germ cell tumours of young adulthood, to 1.000, for extracranial and extragonadal germ cell tumours of childhood and retinoblastoma, with the average of 0.9854. The predicted number of driver events varied from 0.4, for extracranial and extragonadal germ cell tumours of childhood, to 8.95, for malignant gonadal germ cell tumours. The predicted average time between the events varied from 1.4 years, for hepatoblastoma, to 14.85 years, for intracranial and intraspinal embryonal tumours. The predicted maximal populational

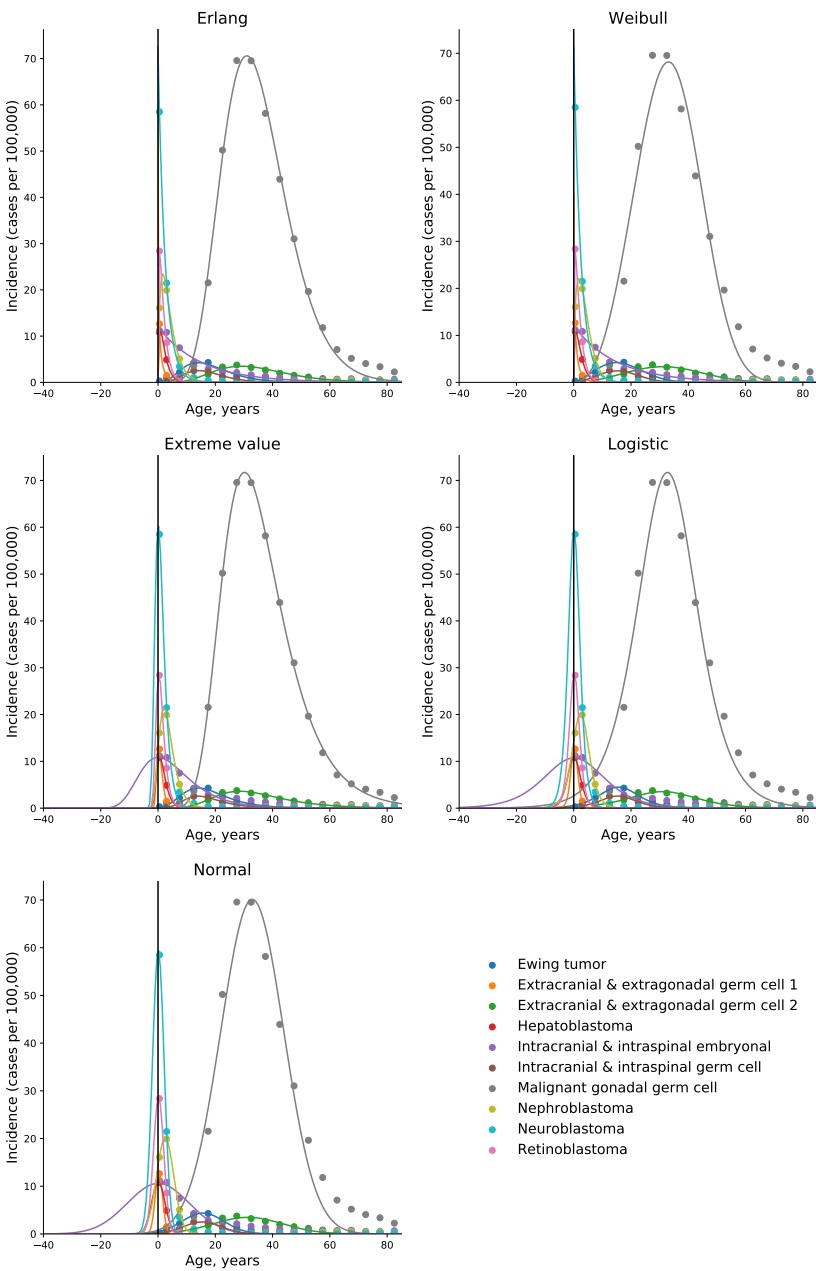

**Figure 2** **Only the gamma/Erlang and the Weibull distributions fit the actual age distributions of childhood and young adulthood cancer incidence without requiring negative age values.** Dots indicate crude incidence rates for 5-year age groups, curves indicate probability density functions fitted to the incidence data for various childhood/young adulthood cancer types. The middle age of each age group is plotted. See Supplemental Information for the optimal parameter values.

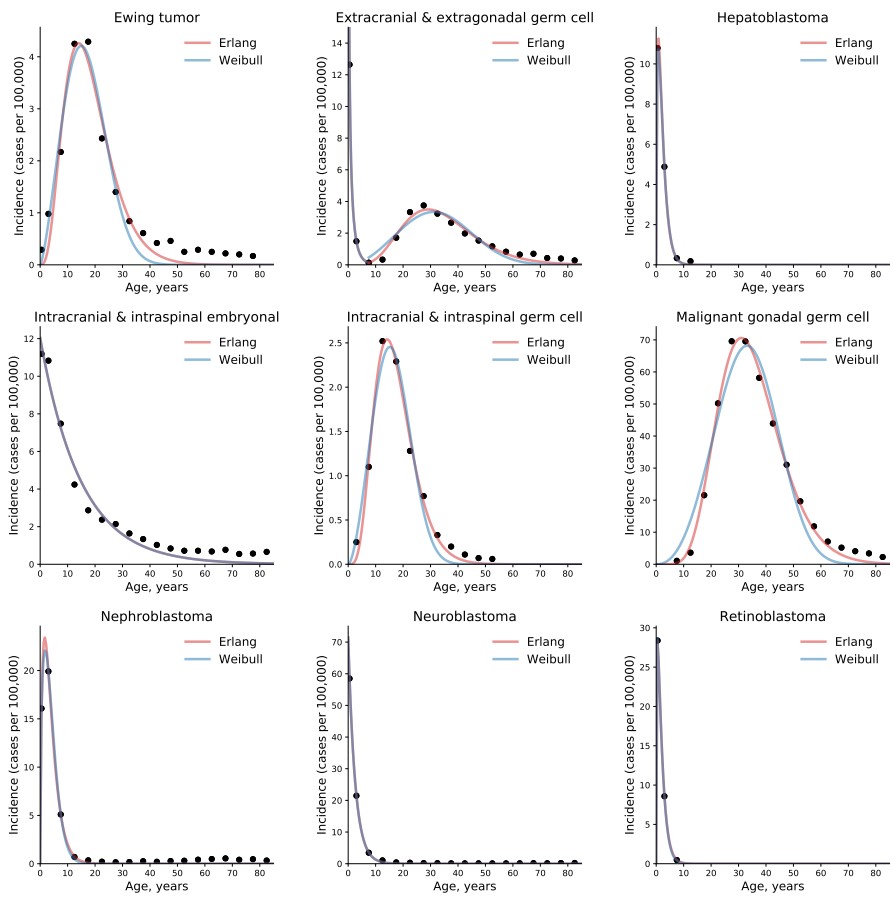

**Figure 3** **The gamma/Erlang distribution approximates the age distribution of incidence for childhood and young adulthood cancers better than the Weibull distribution.** Dots indicate crude incidence rates for 5-year age groups, curves indicate the probability density function of the gamma/Erlang (red) or Weibull (blue) distribution fitted to the incidence data (see Table 2 for $R^2$ comparison and Table 3 for estimated parameters). The middle age of each age group is plotted. Extracranial and extragonadal germ cell tumours of childhood and young adulthood are shown on the same plot.

susceptibility varied from 0.0322%, for extracranial and extragonadal germ cell tumours of childhood, to 1.966%, for malignant gonadal germ cell tumours. Overall, the data show the same high heterogeneity in carcinogenesis patterns revealed in the previous study (*Belikov, 2017*).

## DISCUSSION

We have previously shown that five probability distributions—the extreme value (Gumbel), gamma/Erlang, normal, logistic and Weibull—approximate the age distribution of incidence for the 20 most prevalent cancers of old age (*Belikov, 2017*). The shape of those incidence distributions resembles the bell shape of the normal distribution, with some asymmetry, or at least the left part of it. However, many cancers of childhood have a radically different shape of the incidence distribution, resembling the shape of the exponential

**Table 2  Comparison of the goodness of fit (R²) of the gamma/Erlang and Weibull distributions to the actual age distributions of childhood/young adulthood cancer incidence.** The best fit for each cancer type is highlighted in bold. See Fig. 3 for graphical representation.

| Cancer type | Gamma/Erlang | Weibull |
|---|---|---|
| Ewing tumour and related sarcomas of bone | **0.9516** | 0.9476 |
| Extracranial and extragonadal germ cell tumours of childhood | **1.000** | **1.000** |
| Extracranial and extragonadal germ cell tumours of young adulthood | **0.9495** | 0.8689 |
| Hepatoblastoma | **0.9996** | **0.9996** |
| Intracranial and intraspinal embryonal tumours | **0.9738** | **0.9738** |
| Intracranial and intraspinal germ cell tumours | **0.9890** | 0.9674 |
| Malignant gonadal germ cell tumours | **0.9941** | 0.9530 |
| Nephroblastoma and other nonepithelial renal tumours | **0.9970** | 0.9969 |
| Neuroblastoma and ganglioneuroblastoma | **0.9998** | **0.9998** |
| Retinoblastoma | **1.000** | **1.000** |
| **Average** | **0.9854** | 0.9707 |

**Table 3  Estimated carcinogenesis parameters for 10 childhood/young adulthood cancer types.** The parameters are determined for the gamma/Erlang distribution fitted to actual cancer incidence data (see Fig. 3).

| Cancer type | *k* Number of driver events | *b* Average time between events, years | *A/1000* Maximal populational susceptibility, % |
|---|---|---|---|
| Ewing tumour and related sarcomas of bone | 4.3 | 4.25 | 0.0846 |
| Extracranial and extragonadal germ cell tumours of childhood | 0.4 | 2.35 | 0.0322 |
| Extracranial and extragonadal germ cell tumours of young adulthood | 6.5 | 5.3 | 0.111 |
| Hepatoblastoma | 1.55 | 1.4 | 0.0338 |
| Intracranial and intraspinal embryonal tumour | 1.0 | 14.85 | 0.179 |
| Intracranial and intraspinal germ cell tumours | 5.65 | 3.05 | 0.0426 |
| Malignant gonadal germ cell tumours | 8.95 | 3.9 | 1.966 |
| Nephroblastoma and other nonepithelial renal tumours | 1.75 | 2.2 | 0.124 |
| Neuroblastoma and ganglioneuroblastoma | 1.0 | 2.5 | 0.179 |
| Retinoblastoma | 1.3 | 1.45 | 0.072 |

distribution (Fig. 3). Of the five shortlisted distributions, only the gamma/Erlang and Weibull distributions can assume such shape, when the parameter *k* equals one or less. Other distributions can fit the data only by extending their probability density function into the negative range of age values (Fig. 2), which makes their proper biological interpretation highly unlikely. Of the remaining two distributions, the gamma/Erlang provides superior fit compared to Weibull. In fact, for cancers of old age, the average R² for the Weibull

distribution is 0.9938, whereas for the gamma/Erlang distribution is 0.9954 (*Belikov, 2017*). For cancers of childhood and young adulthood, the average $R^2$ for the Weibull distribution is 0.9707, whereas for the gamma/Erlang distribution is 0.9854 (Table 2). Thus, it appears that the gamma/Erlang distribution is the only classical probability distribution that fits universally well to cancers of childhood, young adulthood and old age. The Weibull distribution can also approximate cancer incidence (*Webster, 2019*), but would only be exactly correct for cancers with $k = 1$, when it becomes the exponential distribution.

We have proposed that the parameter $k$ of the Erlang distribution indicates the average number of driver events that need to occur in order for a cancer to develop to a stage that can be detected during clinical screening (*Belikov, 2017*). The Erlang distribution, like the Weibull distribution, reduces to the exponential distribution when $k$ equals one, because the exponential distribution describes the waiting time for a single random event. It would thus mean that cancers with the exponential shape of the age distribution of incidence require only a single driver event with random time of occurrence, most likely a somatic driver mutation (*Pon & Marra, 2015*) or epimutation (*Roy, Walsh & Chan, 2014*). This explains their maximal prevalence in the early childhood.

In his seminal paper (*Knudson, 1971*), Alfred Knudson has proposed that hereditary retinoblastoma is caused by a single somatic mutation in addition to one heritable mutation. He also proposed that in the nonhereditary form of the disease, both mutations should occur in somatic cells. As hereditary form is estimated to represent about 45% of all cases (*Knudson, 1971*; *Dimaras et al., 2012*), the number of driver mutations predicted from combined incidence data should be around 1.55. Interestingly, whilst the gamma/Erlang distribution fits the incidence data excellently, with $R^2 = 1.0$, it predicts 1.3 driver events (Table 3). This yields the estimate of the hereditary form prevalence at 70%. This higher value may point to the general underestimation of the hereditary component in unilateral retinoblastoma, perhaps due to limitations of routine genetic screening and the influence of genetic mosaicism (*Chen et al., 2014*). In contrast to retinoblastoma, the hereditary form of neuroblastoma is estimated to comprise only 1–2% of all cases (*Tolbert et al., 2017*), hence the exponential age distribution of incidence would mean that only one somatic mutation is required. Indeed, the gamma/Erlang distribution predicts one driver event (Table 3).

The prediction of a single driver event in cancers with the exponential age distribution of incidence does not mean that only a single driver gene can be discovered in such cancer types. In fact, many driver genes are redundant or even mutually exclusive, *e.g.*, when the corresponding proteins are components of the same signalling pathway (*Vandin, Upfal & Raphael, 2012*). Thus, each tumour in such cancer types is expected to have a mutation in one driver gene out of a set of several possible ones, in which all genes most likely encode members of the same pathway or are responsible for the same cellular function. For example, in each neuroblastoma tumour sample, a mutation was present in only one out of five putative driver genes—*ALK*, *ATRX*, *PTPN11*, *MYCN* or *NRAS* (*Pugh et al., 2013*).

Another aspect to consider is that while one mutation is usually sufficient to activate an oncogene, two mutations are typically required to inactivate both alleles of a tumour suppressor gene. Therefore, cancers with the exponential age distribution of incidence are predicted to have either a single somatic mutation in an oncogene, or a single somatic

mutation in a tumour suppressor gene plus an inherited mutation in the same gene. The former is the case for neuroblastoma, where an amplification or an activating point mutation in *ALK* is often present (*Chen et al., 2008*; *George et al., 2008*; *Janoueix-Lerosey et al., 2008*). The latter is the case for retinoblastoma, where an inactivating mutation in one allele of *RB1* is usually inherited, whereas an inactivating mutation in the other *RB1* allele occurs in a somatic cell (*Friend et al., 1986*).

The number of driver events predicted by the Erlang distribution refers exclusively to rate-limiting events responsible for cancer progression. For example, it was shown that inactivation of both alleles of *RB1* leads first to retinoma, a benign tumour with genomic instability that easily transforms to retinoblastoma upon acquiring additional mutations (*Dimaras et al., 2008*). In this case, two mutations in *RB1* are rate-limiting, whereas mutations in other genes are not, because genomic instability allows them to occur very quickly. In neuroblastoma, frequent *MYCN* amplification and chromosome 17q gain are found only in advanced stages of the disease (*Brodeur et al., 1984*; *Bown et al., 1999*), so they are unlikely to be the initiating rate-limiting events.

Application of the gamma/Erlang distribution to childhood and young adulthood cancers showed its exceptionality amongst other probability distributions. The fact that it can successfully describe the radically different age distributions of incidence for cancers of any age and any type suggests that the underlying Poisson process is the universal principle governing cancer development. The Poisson process implies the fundamentally random timing of driver events and their constant average rate (*Belikov, 2017*). Indeed, it has been shown that the number of mutations in cancers (*Tomasetti, Vogelstein & Parmigiani, 2013*) and stem cells (*Blokzijl et al., 2016*) from various tissues increases linearly with age, *i.e.*, with constant rates. The gamma/Erlang distribution allows to estimate, by multiplying the number of driver events by the average time interval between them, that an average person needs from 73 to 324 years to accumulate the required number of driver alterations, depending on the cancer type (*Belikov, 2017*). These estimates correspond to peaks in cancer incidence, because accumulating mutations faster or slower than at an average rate is less likely. Notably, only the susceptible part of the population will develop a given type of cancer, hence the area under the incidence curve never reaches the total population size. Moreover, the finding that driver mutations accumulate for decades is consistent with the silent accumulation of driver mutations in stem cells before the terminal clonal expansion (*Tomasetti & Vogelstein, 2015*; *Blokzijl et al., 2016*; *Gerstung et al., 2020*), because this is the only type of dividing cells surviving for so long in the body, and DNA damage requires cellular division to be fixed in the form of mutations. For childhood and young adulthood cancers, these estimates range from one to 35 years (see Table 3), but the mechanism is likely the same.

Intriguingly, the parameters $k$ (the number of driver events) and $b$ (the average interval between them, in years) demonstrate inverse correlation in adult cancers (see Table 1 in (*Belikov, 2017*)). This could be explained by the observation that all incidence peaks of adult cancers are located at the age of 73 years old or greater, therefore cancer types with high mutation rates (and thus low $b$ values) would accumulate more driver mutations ($k$) by that time than cancer types with low mutation rates (and thus high $b$ values). Although

this answers the initial question it raises a new one: why cancers with high mutation rates do not simply peak at earlier ages and with lower $k$, if such low $k$ is sufficient for low mutation rate cancers? One likely explanation is that an evolutionary pressure of natural selection ensured that cancers do not routinely appear during the reproductive period of life and therefore increased the number of tumour suppressor mechanisms to overcome by high mutation rate cancers. Thus, common cancers can appear only at the old age when evolutionary pressure is absent or minimal. Recently. it has been proposed that age-related degradation of stem cell niche microenvironment is the major reason for positive selection of driver mutations in stem cells and for late-life cancer incidence peaks (*Rozhok & De Gregori, 2015*; *Rozhok & De Gregori, 2019*). That theory, however, has yet to explain the decline in incidence after the peak.

Simulations suggest that the number of human (hematopoietic) stem cells increases from birth until adolescence and then plateaus (*Catlin et al., 2011*). This should lead to the gradual increase in childhood cancer incidence with age, as the target population for mutations grows. On the other hand, modelling suggests that the replication rate of human (hematopoietic) stem cells decreases from ∼17 times in the first year of life, to ∼2.5 times/year between the ages of 3 and 13 years, to ∼0.6 times/year in adults (*Sidorov et al., 2009*). This should lead to gradual decrease in childhood cancer incidence with age, as the stem cell mutation rate heavily depends on the cell division rate. What is the net effect of these two processes with opposing effects on childhood cancer incidence is not clear (*Rozhok & De Gregori, 2015*). Our modelling with the constant driver event rate parameter closely approximates the actual incidence of multiple childhood cancers, which suggests that increases in stem cell pool sizes and decreases in stem cell division rates indeed counterbalance each other and have negligible net effect on childhood cancer incidence.

Another interesting prediction of our model would be that the number of driver mutations per stem cell or per tumour would have a Poisson distribution across the patients' population of any given age. Finally, as the gamma/Erlang distribution allows to predict the number and rate of driver events in any cancer subtype for which the data on the age distribution of incidence are available, it may help to optimize the algorithms for distinguishing between driver and passenger mutations (*Raphael et al., 2014*), leading to the development of more effective targeted therapies.

### Funding
Aleksey V. Belikov received MIPT 5-100 program support for early career researchers. The funders had no role in study design, data collection and analysis, decision to publish, or preparation of the manuscript.

### Grant Disclosures
The following grant information was disclosed by the authors:
MIPT 5-100.

## Competing Interests

The authors declare there are no competing interests.

## Author Contributions

- Aleksey V. Belikov conceived and designed the experiments, performed the experiments, analyzed the data, prepared figures and/or tables, authored or reviewed drafts of the paper, and approved the final draft.
- Alexey Vyatkin performed the experiments, analyzed the data, prepared figures and/or tables, and approved the final draft.
- Sergey V. Leonov conceived and designed the experiments, authored or reviewed drafts of the paper, and approved the final draft.

## Data Availability

The data and scripts are available at GitHub: https://github.com/belikov-av/childhoodcancers.

## Supplemental Information

Supplemental information for this article can be found online at http://dx.doi.org/10.7717/peerj.11976#supplemental-information.

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
