# Peer review of "The Erlang distribution approximates the age distribution of incidence of childhood and young adulthood cancers"

_PeerJ, doi:10.7717/peerj.11976_

## Round 0.1 · original submission · Major Revisions

You will see that both reviewers have raised serious concerns with your manuscript that preclude its acceptance in its current form. Therefore, if you wish to submit a revised manuscript, you will need to substantially revise the paper to take into account each and every one of the comments. A simple rebuttal will not be acceptable.

Thus, you will need to take into account the "non-mutational" aspects of cancer development (we know now that mutation is not equal to cancer, or vice versa - for examples, there is now good evidence that normal cells may acquire 4 or more mutations in cancer-driving genes without becoming cancerous; immune responses and microenvironments are important; etc). The comments on the reality of the estimations and the changing size of the cell of origin of Reviewer 1 must also be addressed, as well as the multiple comments and problems raised by Reviewer 2, detailed in their attached annotated manuscript.

·

Basic reporting

Mostly ok, but needs better referencing of prior work.

Experimental design

Assumptions made in their modeling are overly simplistic, and key parameters like stem cell population size and how it changes with age are not considered.

Validity of the findings

The results of their modeling, such as for numbers of driver events for different cancers, are not substantiated and do not appear consistent with those from cancer sequencing projects.

Additional comments

Belikov and colleagues use various probability models to determine the best fit to the incidences of different childhood and young adult cancers. Understanding the reasons for early life incidences of cancers is clearly important. They present data to claim that that the Erlang/gamma distribution best approximates these incidence curves. From what I can tell, they are exploring two parameters of cancer causation – the occurrence of mutations and their timing. There is no consideration of population size (i.e. for the stem and/or progenitor pools where cancers initiate), mutation rate or the microenvironmental parameters that dictate somatic evolution, nor how these parameters can change quite dramatically with age (from conception to old age). In addition to be overly simplified, I believe this work suffers from over “model fitting”, where one can always find parameters to fit some model, even if the model tells us little to nothing about reality. Moreover, I do not see how their modeling explains the decline in the incidence of these cancers after their early peaks.

Specific concerns:
1) They claim that the Poisson distribution of cancer incidence can be explained by the waiting time for some event (presumably a mutation or epimutation), but why does incidence then fall with the same kinetics that it rose, given that the vast majority of the population remains unaffected? Previous work (from my group) has shown how consideration for how the stem cell pool size changes from conception through childhood can provide at least one reasonable explanation (1, 2). The authors should also consider the work of Mel Greaves on the early risks of childhood leukemias (3, 4).

2) They estimate the numbers of drivers for different cancers of youth; for example, they estimate 8.95 drivers for gonadal germ cell tumors. No attempt is made to link these estimates to the known genetics of these cancers (similarly, their estimates for driver numbers for cancers of old age, in their 2017 Scientific Reports paper, are similarly unsubstantiated).

3) This worked is based on the assumption that “cancers develop upon acquiring a particular number of (epi)mutations in driver genes”. While this is still often accepted, numerous studies have demonstrated that this model ignores key parameters of evolution (like context-dependent selection, for which there are now numerous demonstrations). See for example this review (5).

4) The authors have referenced some classic papers on the age-dependence of cancers, but are missing many others.

5) The goodness of fit often was in the range of R^2 of 0.9 or better, the result of a grid search for combination of parameters that maximized this R^2. Thus, parameters can be found that fit the data, but there’s no indication that these parameters have any basis in reality.

Signed: James DeGregori

1. Rozhok A, DeGregori J. A generalized theory of age-dependent carcinogenesis. eLife. 2019;8:e39950.
2. Rozhok AI, Salstrom JL, DeGregori J. Stochastic modeling reveals an evolutionary mechanism underlying elevated rates of childhood leukemia. Proceedings of the National Academy of Sciences of the United States of America. 2016.
3. Greaves M. Infection, immune responses and the aetiology of childhood leukaemia. Nature reviews Cancer. 2006;6(3):193-203.
4. Greaves M, Cazzaniga V, Ford A. Can we prevent childhood Leukaemia? Leukemia : official journal of the Leukemia Society of America, Leukemia Research Fund, UK. 2021.
5. Solary E, Laplane L. The role of host environment in cancer evolution. Evolutionary applications. 2020;13(7):1756-70.

Reviewer 2 ·

Basic reporting

See attachment for details.

There are places where unsubstantiated claims are made, but I think this may be due to poor English skills.

Experimental design

See attachment for details.

There is a key paragraph that explains the method of fitting the data, that I could not follow. I think this needs improving.

Validity of the findings

See attachment for details.

There is a key paragraph that explains the method of fitting the data, that I could not follow. I think this needs improving.

Additional comments

Overall, I think this is a worthwhile paper, and I hope it can be improved and corrected, for example as suggested in my attachment.

Annotated reviews are not available for download in order to protect the identity of reviewers who chose to remain anonymous.

---

## Round 0.2 · Minor Revisions

Thank you for taking the reviewers' comments into account. Both of the reviewers indicate that the revised manuscript is potentially acceptable for publication after some minor revisions, as indicated in their comments and in the attached annotated manuscript.

·

Basic reporting

The paper is well written (there are a few grammatical mistakes, but minor). Referencing is more balanced in the revised manuscript.

Experimental design

The authors now better acknowledge the role of the tissue microenvironment (basically, in terms of how it influences the "driverness" of a mutation). I still think that they should consider, or at least acknowledge, that the population size of stem cells, which obviously dramatically changes from conception to maturity, will influence cancer rates - essentially, the target population for mutations is changing. In addition, tissue microenvironments change (particularly in old age). That said, I do not expect that every model of cancer will focus on the same parameters that we find important, so I do not expect the authors to change their modeling methods. But simply acknowledging that parameter changes with age should influence cancer rates, even if not incorporated into their models, would increase the balanced nature of the work.

Validity of the findings

The modeling appears sound (noting that I do not have the mathematical expertise to judge their methods, although Rev #2 appears to). They now better indicate that their modeling shows what "could be" not what "is".

Additional comments

I think that it's important for science that different views are encouraged even when they disagree with the views and results of the reviewer (me). The authors did do a good job addressing concerns, with the minor exceptions noted above (particularly for stem cell pool size changes with age). In all, their work does make a meaningful contribution to the overall debate for how cancer incidence varies as a function of age. I am choosing "minor revisions" simply to provide the authors with the opportunity to acknowledge the shortcomings of their modeling in terms of considerations of population size (stem cells) and tissue microenvironments and how these CHANGE with age.

Reviewer 2 ·

Basic reporting

This is now much clearer. I have suggested a handful of minor changes to the text to improve the language - see attachment.

Experimental design

I have not checked the code, but I have no reason to doubt its validity. It was unclear what the data was that is in the repository with the code. It appears to be some form of summary data. An improved description of that data is needed.

Validity of the findings

As far as I can tell, the findings seem valid.

Additional comments

I think the article is much better, I have suggested a small number of minor alterations to improve the text and explain things better.

Annotated reviews are not available for download in order to protect the identity of reviewers who chose to remain anonymous.

---

## Round 0.3 · accepted · Accept

Many thanks for submitting your revised manuscript and for appropriately addressing the previous concerns in your revision.